# The Effect of Human Settlement Pedestrian Environment on Gait of Older People: An Umbrella Review

**DOI:** 10.3390/ijerph20021567

**Published:** 2023-01-14

**Authors:** Changzheng Xuan, Bo Zhang, Xiaohu Jia

**Affiliations:** 1Architecture College, Inner Mongolia University of Technology (IMUT), Hohhot 010051, China; 2Inner Mongolia Key Laboratory of Green Building, Hohhot 010051, China

**Keywords:** human settlements, pedestrian environment, older people, healthy aging, gait, umbrella review

## Abstract

Older people are limited by the pedestrian environment in human settlements and are prone to travel difficulties, falls, and stumbles. Furthermore, we still lack systematic knowledge of the pedestrian environment affecting the gait of older people. The purpose of this review is to synthesize current evidence of effective human settlement pedestrian environments interfering with gait in older people. The systematic effects of the human settlement pedestrian environment on gait in older people are discussed. Databases such as Web of Science, Medline (via PubMed), Scopus, and Embase were searched for relevant studies up to June 2022. The literature was screened to extract relevant evidence from the included literature, assess the quality of the evidence, and analyze the systematic effects of the pedestrian environment on gait in older people. From the 4297 studies identified in the initial search, 11 systematic reviews or meta-analysis studies were screened, from which 18 environmental factors and 60 gait changes were extracted. After removing duplicate elements and merging synonymous features, a total of 53 relationships between environmental factors and gait change in older people were extracted: the main human settlement pedestrian environmental factors affecting gait change in older people in existing studies were indoor and outdoor stairs/steps, uneven and irregular ground, obstacles, walking path turns, vibration interventions, mechanical perturbation during gait, and auditory sound cues. Under the influence of these factors, older people may experience changes in the degree of cautiousness and conservatism of gait and stability, and their body posture performance and control, and muscle activation may also be affected. Factors such as ground texture or material, mechanical perturbations during gait, and vibration interventions stimulate older people’s understanding and perception of their environment, but there is controversy over the results of specific gait parameters. The results support that human settlements’ pedestrian environment affects the gait changes of older people in a positive or negative way. This review may likely contribute evidence-based information to aid communication among practitioners in public health, healthcare, and environmental construction. The above findings are expected to provide useful preference for associated interdisciplinary researchers to understand the interactions among pedestrian environments, human behavior, and physiological characteristics.

## 1. Introduction

Population aging will be one of the most significant social changes of the 21st century [1]. The challenges posed by population aging are wide-ranging [2,3]. To promote healthy and active aging, human settlements should improve their strengths in maintaining health, long-term care, and welfare systems. The functional capacity of older people decreases with age. Functional capacity is the possession of the ability to enable all people to be and do what they have reason to value, and it consists of an individual’s intrinsic abilities, relevant environmental characteristics, and the interactions between them [4]. Walking is an athletic function that significantly affects the level of independence in older people today; the pedestrian environment as a spatial carrier of the walking behavior of older people has attracted the attention of researchers.

Previous studies have shown that the human settlements’ pedestrian environment can have an impact on functional walking performance and motor health in older people. For example, Van Cauwenberg et al. [5] noted a negative relationship between the objectively measured presence of slopes and/or stairs and the utility of a street section for foot traffic. Haselwandter et al. [6] reported that older persons perceived the presence of outdoor stairs as a barrier to walking. The relationship between the built environment and physical activity in older people is seen as a more interactive relationship [7]. On the one hand, with walking as the most common type of physical activity, a good pedestrian environment can promote walking behavior in older people, which is important for their health. On the other hand, older people are more restricted in their mobility in the environment of the human settlement. Built environments such as physical barriers interact with chronic diseases or mobility restrictions [5], affecting the walking ability and health of older people. The design of adapted human settlements’ pedestrian environments for older people is a key issue today.

Current human settlements’ pedestrian environments do not provide suitable universal designs to meet the needs of older people. A large number of steps, ramps, obstacles, and other factors in the pedestrian environment in human settlements bring challenges to the walking of older people. The large variability and randomness in the design of the pedestrian environment in human settlements also affect the walking experience of older people [7] and have implications for their access to social resources and daily travel. Changes in the pedestrian environment also make gait and posture maintenance more difficult for older people and bring about a range of gait changes that can be used to characterize the quality of older people’s gait in relation to the human settlements’ pedestrian environment.

The effect of pedestrian environment on walking in older people can be demonstrated by gait parameters. Studies suggest that aging has been associated with modifications in spatiotemporal [8] and kinematic [9] gait parameters. For example, older people showed a decline in age-related sensory systems, and the ability to adapt to environmental changes to maintain balance becomes low. Spatiotemporal gait changes with increasing age in older people, such as slower gait and increased gait variability, and these parameters are amplified with exposure to visual disturbances [10]. In this, human gait encompasses the quantification and interpretation of measurable walking gait parameters. Gait conventionally involves the body motion (kinematics) and boundary conditions (ground reaction forces (GRF(t)) and moments (GRM(t)) or alternatively GRF(t) and the plantar center of plantar pressure (COP(t)) under each foot). The joint forces and torques (kinetics) and muscle forces can subsequently be estimated from these measured data. The spatiotemporal gait parameters, such as stride length, joint angles, peak GRF(t), and inter-segmental forces, provide important measures of the health and performance of the gait [11]. These gait parameters can be used to demonstrate the relationship between physical function and kinematics in older people, but related studies have mostly focused on physiological health aspects and less on the effects between the pedestrian environment and gait parameters.

Some studies address changes in the environment [12,13,14,15]. Researchers and practitioners alike have come to appreciate the importance of the built environment in facilitating or constraining walking. Walking occurs primarily in neighborhood streets and public facilities, and the character of such places influences the degree to which they are safe, comfortable, and attractive for walking [16]. In order to create places that facilitate and encourage walking, practitioners need an understanding of the specific characteristics of the built environment that correlates most strongly with walking. Some authors use different categories to classify the elements of the built environment, splitting out the features of the physical environment in land-use patterns, urban design characteristics, and transportation systems [17], not categorizing the elements [18] or considering only safety domains of the neighborhood [19]. Other authors [20] classify physical environmental variables according to Neighborhood Environmental Walkability Scale (NEWS), which includes the category “walkability” as the main and most generic attribute. However, most of these studies have focused on community walkability characteristics at the meso-/micro-scale. Studies of geographic measurements or conventional walkability may not capture local variation in environmental characteristics and have paid less attention to the specific effects of micro-scale environments on gait parameters in older people.

A growing number of studies have also explored the role of the environment in the regulation of gait in older people [8,9,21,22]. Several design criteria and ergonomic studies have analyzed the relationship between human settlements’ pedestrian environments and kinematic measures [22,23,24,25], conducted systematic reviews or meta-analyses of the effects of certain factors in the pedestrian environment of older people, and examined the effects of factors such as stairs, obstacles, and ground conditions of the pedestrian environment on the gait of older people. However, these studies only focus on local factors of the pedestrian environment, without the necessary systematic elaboration of the overall pedestrian environment, and fail to provide systematic evidence of the effect of the human settlements’ pedestrian environment on the gait of older people. There is a lack of an umbrella review outlining all the systematic reviews published in areas related to the broad range of pedestrian environments affecting gait in older people. The aim of this review is to synthesize current systematic review and meta-analyses evidence on the efficacy of human settlements’ pedestrian environment intervention in older people’s gait. To present an overview of the systematic effects of walking in human settlements’ pedestrian environment on the gait characteristics of older people.

## 2. Methods

### 2.1. Design and Registration

An umbrella review is a review method based on systematic review and meta-analysis [26]. This umbrella review is a systematic collection and assessment of multiple systematic reviews [27]. The study protocol was developed in accordance with the reporting guidance in the Preferred Reporting Items for Systematic Reviews and Meta-Analyses Protocols (PRISMA-P) [28] statement and is registered in the International Prospective Register of Systematic Reviews (PROSPERO: CRD42022364172).

### 2.2. Data Sources and Search Strategy

We systematically searched the Web of Science, MEDLINE (via PubMed), Scopus, and Embase databases using a comprehensive search strategy up to June 2022, in addition to reference lists including systematic reviews to identify potentially eligible studies. Search terms constructed coverage for umbrella reviews according to the PICOS framework. The searches will be developed and combined using broad search terms, keywords, and MeSH terms. To be deemed relevant for review, an article had to contain words referring to walking, older people, gait, and an environment obstacle in the title, abstract, and keywords. Due to differences in terminology and writing styles among researchers, a list of synonyms was used for each key term to identify all relevant articles. Key terms within the search strategy were matched and exploded using medical subject headings (MeSH) in databases such as MEDLINE. For details of the keywords and the systematic search strategy, see Appendix A Table A1.

### 2.3. Inclusion and Exclusion Criteria

The PICOT (population, intervention, comparison, outcome, type of study) (see Table 1) [29] scheme was used to identify the papers that met the inclusion criteria. The reviewers conducted a preliminary screening of the titles and abstracts, and articles unrelated to walking were excluded. The remaining abstracts were checked against the inclusion and exclusion criteria. The full text of an article was examined if there was insufficient information contained in the abstract to decide whether to include the article. The references section of queried articles was surveyed to identify additional relevant studies. Articles that are related to animal studies, robotics, pulmonary obstruction, children, osteoarthritis, and amputees were excluded. The participants in the primary studies selected for the systematic reviews included here were older people without definite physical illness. Published studies of subjects with diagnosed gait and/or equilibrium disturbance were excluded. Potential deterioration associated with aging may be a cause of gait changes. This paper does not exclude such study participants who may have potential deterioration in order to focus on the normal consequences of aging results in older people. In this paper, the pedestrian environmental factors of human settlement were used as the primary intervention, and the control group chosen was determined to be the baseline walking status of older people when they were not subjected to one or more environmental interventions, such that the results obtained were the effects experienced by older people. Articles comparing with young people of different age groups were excluded because they focused on constant senescence as a main variable condition. Two reviewers (C.X., B.Z.) independently selected the titles, abstracts, and full texts according to the inclusion and exclusion criteria. Any disagreements were solved by consensus and, if disagreement persisted, by a third reviewer (X.J.). We included all systematic reviews that contained intervention studies (e.g., randomized control trials (RCT), cluster-randomized studies, cross-over studies, or quasi-randomized studies, as well as single-arm intervention studies with before/after comparisons as primary studies) and all studies that included laboratory conditions and those conducted under natural conditions, were included in this paper to ensure that this paper provides comprehensive evidence-based results.

### 2.4. Methodological Quality

The methodological quality of the included systematic reviews was assessed using the A MeaSurement Tool to Assess Systematic Reviews 2 (AMSTAR-2) [30]. This tool was developed specifically to assess the methodological quality of systematic reviews of randomized and non-randomized studies. It has 16 items in total (compared with 11 in the original), has simpler response categories than the original AMSTAR [31], includes a more comprehensive user guide, and has an overall rating based on weaknesses in critical domains. Seven domains (items 2, 4, 7, 9, 11, 13, 15) can critically affect the validity of a review and its conclusions are regarded as weaknesses. AMSTAR 2 does not have an overall score. The methodological quality was usually categorized as high (no or one non-critical weakness), moderate (more than one non-critical weakness), low (one critical flaw with or without non-critical weaknesses), and critically low (more than one critical flaw with or without non-critical weaknesses) [30]. Two authors (C.X., B.Z.) independently assessed the AMSTAR 2, any disagreements were solved by consensus and by a third reviewer (X.J.) if disagreement persisted.

### 2.5. Data Extraction

Using the Cochrane guidance [32], a data extraction form was created regarding key elements of the review methodology (see Table 2). Data were extracted independently by the three authors. After evaluating the articles, the three authors met, discussed, and resolved any discrepancies which were largely related to more subjective criteria such as whether the research question was a good ‘fit’ for a review. Independently extracted information from eligible primary studies of the selected systematic reviews or meta-analyses, the extracted data for each review included:First author, year of publication, search date, number of primary studies meeting the requirements of this umbrella review, the total number of primary studies included in the included systematic reviews, characteristics of the population included (e.g., number of participants, healthy condition, the gender proportion, and age range/mean age).Aim of the review, type of intervention, interventions, and diagnostic criteria.Effect size results for intervention controls, metric of effect size.Overall results and main findings, overall recommendations, and limitations as provided by the review itself.

### 2.6. Data Synthesis and Analysis

Given the heterogeneity of populations, outcomes, and analyses, the results of the included reviews were summarized primarily using narrative synthesis and tabulated and visualized in conjunction with appropriate quality scores. The primary analysis of this umbrella review is centered on intervention type. Thereafter, cross-sectional comparisons of similarities and differences under different interventions are reported and discussed. The outcomes of each systematic review of inclusions were considered and discussed in the context of their methodological quality, as determined by the AMSTAR 2 and the GRADE algorithm [12,33,34,35,36].

### 2.7. Evidence Map

We present the scientific evidence for each systematic review through an evidence map. We present the information for each review using the following criteria:Number of studies (figure size): the size of each figure is directly proportional to the number of original studies included in each of the meta-analyses.Study quality and type of action relationship (symbol): The methodological quality evaluated in each study is represented by color. The action relationship type determines each symbol inside the bubble (X: contested effect relationship; −: negative correlation; +: positive correlation).*X*-axis: in the descriptive mapping, the environmental variables of each study are on the X-axis.*Y*-axis: descriptive mapping representing the gait characteristics variables of the older people for each study.

**Table 2 ijerph-20-01567-t002:** Basic features of studies.

References, Author(s) (year)	Search Date	Included in the Umbrella Review/ Included in the Review	Aim of the Review	Number of Participants	Health Condition	Age	Intervention	+ *	−	Gait Effect
Mehdizadeh et al. (2021) [37]	20 February	4/34	Investigate the association between measures of gait center of pressure (COP) and aging and falls.	65	Healthy older people	Over 65	During obstacle crossing		√	Average anterior-posterior and medio-lateral center of pressure displacements
									√	Average anterior-posterior and medio-lateral center of pressure velocity
Valipoor et al. (2020) [38]	2018	4/45	To determine the current state of knowledge of the effects of internal scale elements of the human settlements’ environment on falls in older people, the extent to which the effects are studied, and the extent to which these effects are measured.	95	Older people	/	Carpet	/		Postural sway
							Flooring type	/		Dynamic balance
Ghai et al. (2017) [39]	17 May	18/34	To analyze the effects of rhythmic auditory cues on spatiotemporal gait parameters among healthy young and older people participants.	355	Healthy older people	68 ± 5.6 years	Rhythmic auditory cues	√		Gait velocity
								√		Stride length
								√		Cadence
									√	Coefficient of variability stride time
									√	Coefficient of variability stride length
							Non-modulated rhythmic auditory cues	√		Gait velocity
								√		Stride length
								√		Cadence
Aboutorabi et al. (2017) [40]	2002 to July 2015	12/15	To summarize the current evidence for subthreshold vibration interventions on postural control and gait in older people.	250	Health older people and other	Over 55	Vibration interventions		√	Postural sway
								√		Postural stability
									√	COP velocity
								√		Balance
								√		Gait speed
								√		Cadence
							Noise stimulation	/		Stride length
								/		Gait variability
Uiga et al. (2015) [41]	July 2014	11/25	To synthesize the available evidence on the role of gaze behavior during locomotion (i.e., walking, turning,) to explore the role of gaze behavior among older people during different forms of locomotion.	437	Health older people and other	Over 60	Stepping over an obstacle		√	Stride length
									√	Gait velocity
								√		Saccade/step latency
							Walking up and down stairs	√		Saccade/step interactions time
									√	Stride length
								√		Single stance phase
							Turning around a corner		√	Gait velocity
Nightingale et al. (2014) [42]	February 2012	24/88	Summarize the literature related to timed stair tests. We aimed to determine whether stairs would be able to differentiate between participant groups by age or impairment.	3365	Healthy older people	Over 65	Stair ascent		√	Gait velocity
							Stair descent		√	Gait velocity
							Stair ascent/stair descent		√	Gait velocity
Orth et al. (2013) [43]	August 2012	7/23	This study was to evaluate the efficacy of textured materials for enhancing perceptual motor functionality.	/	Healthy older people	Over 65	Textured materials	/		Gait performance
Alfuth et al. (2012) [44]	2000 to 2011	7/23	This paper was to provide a review of studies reporting the effects of changes in plantar feedback on human gait characteristics.	307	Health older people and other	Over 55	Walking on uneven surface and irregular surface		√	Gait velocity
									√	Step length
									√	Step width
									√	Cadence
								√		Gait variability
Galna et al. (2009) [45]	August 2008	11/15	The primary aim was to investigate whether older people are more likely to contact ground-based obstacles when walking under time-constrained and unconstrained conditions. The second aim examined whether older people approach and step over obstacles using different spatiotemporal, kinematic, and kinetic strategies than younger adults.	/	Healthy older people	68–76	During obstacle crossing		√	Stride length
									√	Gait velocity
									√	Measure maximum range of motion (rom)
								/		Hip, knee, and ankle range of motion
								√		Eccentric contraction of the stance limb hip abductors
									√	Vertical hip power (defined as the power used to raise the stance hip) at toe-off
									√	Pelvic drop during stance, the height of the crossing limb.
Lachance et al. (2017) [46]	October 2015	14/84	What evidence exists from experiments conducted in a controlled, laboratory environment about balance, gait and mobility performance, and/or assistive device use on compliant flooring systems.	/	Older people	Over 65	Compliant flooring systems	/		Gait speed
								/		Step length
								/		Toe clearance
								/		Timed up-and-go time
Taylor et al. (2022) [10]	2015 to 30 May 2022	12/33	This systematic review aimed to assess current perturbation methods and outcome variables used to report participant biomechanical responses during walking.	294/779	Healthy older people	Over 65	Perturbations during gait	√		In base of support
								√		Muscle activation
								√		Dynamic stability
								√		Margin of stability
									√	Centre of mass sway
								√		Cadence
									√	Single support time and swing time
								/		Joint angle range of motion,
								/		Step width
								/		Step length

* −, negative correlation; +, positive correlation; /, Correlation unclear or disputed.

## 3. Results

### 3.1. Search Results

Of the 4297 studies initially located and downloaded, 1221 were removed due to duplication. During title and abstract screening, an additional 2975 studies that did not meet the inclusion criteria were excluded. A reference search of the remaining literature identified 159 additional articles that met the inclusion criteria, leaving a total of 260 studies for full-text screening. Of these articles, 249 were excluded because they did not meet the inclusion criteria. This resulted in a total of 11 systematic reviews or meta-analyses which were included in this umbrella review [10,37,38,39,40,41,42,43,44,45,46] (for more details see the flowchart in Figure 1).

### 3.2. Characteristics of Included Studies

This umbrella review selected 11 articles that met the inclusion criteria, Table 2 provides details of the interventions and results of each study and key results. A wide range of kinetic, kinematic, spatiotemporal, and muscle activity outcome variables was reported. The included systematic review was published between 2009 and 2022. Among these systematic reviews, 2 were meta-analyzed [39,43], and the remaining 9 introduced their findings through narrative synthesis [10,37,38,40,41,42,44,45,46]. The 11 systematic reviews included 419 primary studies. As the subjects of these studies are not only older people, 9 of them were realized from the subgroups in the study. When the irrelevant subgroups were excluded, 120 primary studies on the gait characteristics of older people from environmental factors were found. Among them, except that 3 systematic reviews did not provide the number of samples included in the study [43,45,46], the remaining 8 systematic reviews adopted a total of 5168 samples. Due to the different definitions of older people in the world, except for 1 study that does not provide the age characteristics of the older people under study [38], there are 2 other studies in which the standard for the use of systematic review is older than 55 years old [40,44], and the standard for the use of 8 systematic reviews is older than 65 years old [10,37,39,41,42,43,45,46]. Most of the participants in the primary study were older people without definite physical illness. Potential aging-related deterioration may be one of the causes of gait changes. Some studies did not distinguish the specific physical conditions of the research subjects [38,40,41,44]. The heterogeneity of the research results brought about by the physical health of older people may affect the interpretation of the research results of this umbrella review.

### 3.3. Methodological Quality of Included Studies

The methodological quality of the included systematic review is shown in Figure 2. The quality included in the review ranged from very low to medium, and none of the studies included in the review were rated as high quality. Most of the comments were rated as extremely low [37,38,42,44,45,46], two systematic comments were rated as low [40,41] and three comments were rated as medium [10,39,43]. Among the key indicators, only a few systematic review research methods have been determined before implementation [10,46], and most of the studies did not use reasonable tools to assess the bias risk of the included research literature [37,38,41,42,44,45,46], which significantly affected the quality of the included systematic review and meta-analysis methods. Among non-key indicators, due to the different types of studies selected for inclusion in the systematic review, some studies did not explain the reasons for selecting the type of study design included in the system review [10,37,41,42,43,44,45,46]. Most of the studies rated as low or extremely low were considered to have deficiencies related to prior protocol development, comprehensive search strategies, and exclusion lists and reasons.

### 3.4. Findings: Associations between Gait Outcomes and Environment Features

In this paper, a total of 17 environmental factors and 60 related gait changes were extracted from 11 systematic reviews that met the criteria. After removing duplicate elements and merging synonymous features, a total of 9 environmental factors and 23 gait changes closely related to them were extracted, and there were a total of 53 relationships between the role of environmental factors and gait changes of the older people. Among them, environmental factors mainly refer to the environmental features in the pedestrian environment of human settlements that directly act on older people or indirectly influence their walking behavior. The gait outcomes, as characteristic values of the effect of environmental interventions, were mainly gait time parameters, distance parameters, time-space parameters, and kinetic parameters. Table 3 summarizes the effects between the environmental characteristics of walking included in the review and the gait characteristics of older people. Figure 3 shows the results of the evidence map presented in conjunction with information systems such as the methodological quality of the studies and the total number of primary studies accepted for systematic review.

#### 3.4.1. Stairs or Indoor and Outdoor Steps

Two studies included in this article suggest that stairs-climbing will reduce the step length [41], gait cadence [41], and gait speed [42] of older people, increased the interaction saccade/stepping interactions time and lower limb swing time when the older people walk [41]. The systematic review of Uiga et al. [41] examined the visual guidance during stair climbing by focusing on the saccade/step interaction of older people. They found that older people displayed significantly longer duration between the onset of saccade and the onset of foot-lift up to the platform (i.e., saccade–step latency) and that older people showed significantly lower cadence and longer single-stand phases as they went up and down the stairs. Nightingale et al. [42] found through a systematic review that in healthy older people, the speed of going up and down stairs decreases with age, and the time spent on going down stairs increases more. From the data obtained, it was shown that stair walking time varies with age and can be used to differentiate between different patient groups.

#### 3.4.2. Obstacle

Two studies included in this article suggest that when older people encounter obstacles that need to be crossed during walking, they will reduce the step length and gait velocity [41,45], and increase the time delay of scanning step interaction [41]. Furthermore, one study showed that encountering obstacles while walking affects the center of pressure displacement and speed in older people [37].

A systematic review of 13 studies by Uiga et al. [41] found that older people needed a relatively long saccade/step latency and longer saccade/stepping interaction time before crossing a barrier [47]. Given online changes in the location of multiple stepping targets, older people manifested longer saccadic latency and greater error following medial and lateral changes in target location than young adults [48]. Galna et al. [45] found that the gait of older people is more conservative when crossing the walking obstacle, in the eight that reported differences, older people walked slower during obstacle crossing [49,50,51,52,53,54], with smaller steps [49,50,53,55]. Older people are more likely to take more steps to cross barriers [49] and use small-step strategies [55].

Mehdizadeh et al. [37] found that the average displacement and velocity of anterior–posterior (AP) and medio-lateral (ML) COP were significantly lower in older people when crossing obstacles. In normal walking, older people showed a more intermediate COP trajectory. The COP speed was faster in the initial foot contact, and the COP speed was slower in the middle of the gait cycle. In both gait initiation and obstacle crossing, older people had lower average AP and ML COP displacement.

A study by Galna et al. [45] also found that when older people are faced with obstacles in the pedestrian environment, vertical hip power at toe-off, the height of the crossing limb, and the measured maximum range of motion (ROM) will be reduced, and the eccentric contraction of the hip abductor of the standing limb will be reduced. In the included research, it is found that the forelimbs of older people are closer to the obstacles after crossing the obstacles [50,53]. They demonstrated greater hip flexion during the swing phase of gait for the lead and trail limbs as well as greater. There is also evidence that the internal torque of the hips and ankles is reduced during key activities in the gait cycle of older people crossing obstacles.

#### 3.4.3. Irregular or Uneven Ground Surfaces

In a systematic review by Alfuth et al. [44], a two-fold effect of irregular uneven ground on gait in older people was found. On the one hand, there was a significant correlation between gait and plantar tactile sensation, with walking on uneven ground surfaces leading to a reduction in gait speed, step speed, and step length. When walking on irregular ground surfaces, subjects were significantly slower and took shorter steps. Compared with uneven ground, the subjects walk faster on flat ground [56]. On the other hand, they believe that walking on irregular, uneven ground increases gait variability in older people. In particular, older women show greater stride variability and gait time variability when walking on irregular ground surfaces [57]. Plantar cutaneous mechanoreceptors were considered to give information about the shift of the center of mass and the association between the center of mass and the base of support to regulate the accuracy and position of foot placement during gait termination [44].

#### 3.4.4. Path Turns during Walking

Path turns during walking are considered common in pedestrian environments, and a systematic review by Uiga et al. [41], included in this paper, concluded that turns during walking also reduce gait speed in older people. During turns, the older people were found to be highly dependent on central vision, and the difference in lateral eye movement was significantly higher than that in the younger group. Older people were found to initiate whole-body rotation in tandem with the beginning of a saccade to the direction of the turn, followed by a head, trunk, and feet reorientation, and spend more time gazing at the ground within the first two steps of the obstacle [58]. However, if the direction of the turn was indicated shortly before the turn, older people initiated the segment reorientation via trunk yaw, followed by rapid shifts of gaze, and then medio-lateral feet displacement. Both path turns with and without cues were found to have slower walking speeds and longer turn times in older people [59]. Older people tend to rely more on central vision than peripheral vision and appear to rotate their vision in order to gain greater stability.

#### 3.4.5. Mechanical Perturbation during Gait

Mechanical disturbance during gait means that participants are disturbed by the mechanical balance in gait, the general disturbance sources are escalators, treadmills, and moving ground. In the systematic review of Taylor et al. [10] included in this paper, the common outcome variables of mechanical perturbation during gait are the stride and step size of participants. Some primary studies also found that anterior and lateral split belt treadmill translations [60], hip abductor muscle vibration [61], and differing split belt speeds [62] all showed decreased single support time. Step frequency was agreed to a significant increase following perturbation by treadmill translation [63] or split belt accelerations [62]. Taylor et al. reported contradicting results of participants’ step width, showing an increase following ML treadmill translation [64,65], AP and ML treadmill translations [63], single belt AP and ML treadmill translation [66], split belt treadmill accelerations [61], and hip abductor muscle vibration [60] during perturbed gait. Contradicting this, anterior and lateral split belt treadmill translations [60], gait post vibration of hip abductor muscles perturbation [61], and medial treadmill translations [67,68] were all shown to reduce step width.

Taylor et al. [10] also showed that mechanical gait disturbance decreases the degree of sway of the center of gravity and increases dynamic stability. Six studies included in the systematic review reported measurements of dynamic stability [60,63,68,69,70,71]. AP and ML treadmill belt translations of 2.5 m s^−1^ [63], AP and ML waist belt pulls and pushes [35] and medial belt translation [67], and narrowed step width [71] all reportedly increased participants’ dynamic stability. Two studies reported decreased dynamic stability in response to perturbation [60,68]. Center of mass (COM) sway velocity, sway area, and mean sway during gait was reported by one study [72]. Seven studies reported on the margin of stability (MoS) [69,73,74,75,76,77,78] and one reported on the base of support (BoS) area [73]. All studies reported an increase in MoS.

Studies have shown that mechanical perturbations during gait alter the degree of muscle activation in older people. Four studies reviewed by Taylor et al. reported muscle activation [76,79,80,81]. Gluteus medius activation increased significantly as a result of perturbation application [79]. The tibialis anterior, peroneus longus, and soleus activity were all reported to increase in response to gait perturbation [80], as did mean spectrum power [76]. However, the magnitude of activity in the first principal components of quadriceps, hamstrings, and gastrocnemius activity decreased following overground floor translations [81]. However, the included studies show that the specific impact on the range of motion of the joint angle in older people is still unclear [71].

#### 3.4.6. Vibration Interventions

In this paper, Aboutorabi et al. [40] studied the current evidence for subthreshold vibration interventions on postural control and gait in older people, it shows that vibration interventions can increase the gait velocity and cadence of older people, reduce the postural sway and increase posture stability. In studies reporting the effects of vibration on gait parameters [82,83,84,85,86], In two studies [82,85], mean walking speed was not changed significantly with mechanical noise in healthy older subjects (without vibration 122.9 cm/s, with vibration 124.6 cm/s) [82,85]. However, in other studies, there was an increase in walking speed and cadence in healthy older subjects (velocity improved from 1.25 to 1.32 m/s and cadence improved to 112 steps/min) [83,84,86]. Individual parameters did not change significantly in healthy older people [82,85]. With the exception of three studies [84,85,87], other studies evaluated the effect of vibration stimulation immediately after or during the test. In general, the application of a vibration noise program can improve walking speed and step length. However, consistent results could not be reached for the effects on gait length and gait variability in older people when the noise variation of vibration could not reach the interference threshold.

Most studies included by Aboutorabi et al. [40] also showed that the application of subliminal mechanical noise to the subject’s foot resulted in enhanced feedback and reduced sway parameters in healthy older people [85,88,89,90,91,92]. Although no statistically significant difference was observed in a study by Priplata et al. [93], a reduction in all of the postural sway variables was observed during applying 90% of sensory threshold noise on vibrating insoles in the healthy older people participants [88]. The average area of postural sway with eyes open (EO) and eyes closed (EC) was reduced significantly in five studies by vibratory stimulus in older people subjects (without vibration 190.8 mm^2^, with vibration 159.9 mm^2^) [85,87,91] and greater improvements in the postural sway displacement with EO (from 70.86 to 58.95 cm) and EC (from 91.97 to 71.31 cm) were observed with vibration stimulus [84,90]. Older people fallers showed statistically significant differences between the control and vibration conditions in the AP direction (decreased from 1.43 to 1.32 cm) but older people non-fallers showed no statistically significant difference using the vibratory insoles. In healthy subjects with the use of noise, the COP range decreased from 20.7 to 18.9 mm in the AP direction and from 13.1 to 12.7 mm in the ML direction [92].

#### 3.4.7. Auditory Cues

A systematic review by Ghai et al. [39] included in this paper found that both rhythmic and non-modulated rhythmic auditory cues can significantly increase the step length, gait velocity, and gait cadence of older people. Their meta-analysis found that gait speed, stride length, and cadence were enhanced after rhythmic auditory cues were applied to older participants. In the included thirty-four studies, thirty studies reported significant enhancements, two studies reported enhancements (*p* > 0.05) [94,95], and one study reported a significant reduction in gait parameters with rhythmic auditory cues [96]. The analysis for old participants performing gait with rhythmic auditory cues revealed beneficial effects with medium effect and substantial heterogeneity (g: 0.68, 95% C.I: 0.28 to 1, I^2^: 81%, *p* < 0.01). Further, sub-group analysis with non-modulated rhythmic auditory cues revealed, under a single task condition, revealed a medium effect size with substantial heterogeneity (Hedge’s g: 0.73, 95% CI: 0.2 to 1.2, I^2^: 80.2%, *p* < 0.01). Their systematic review also showed that rhythmic auditory cues can significantly improve the gait stability of older people [39]. Out of thirty-four included studies, 88% studies reported beneficial effects of rhythmic auditory cues on primary spatiotemporal gait parameters. The meta-analysis also reported a decrease in stride time and length variability coefficients in older participants with rhythmic auditory cues [39].

#### 3.4.8. Textured Materials

The systematic review of Orth et al. [43] included in this paper found that the effects of textured material stimulation on gait performance in older people remain controversial. Their systematic review observed the effect of increased texture during walking, and a variety of gait parameters were reported including spatiotemporal (walking velocity (m/s), cadence (steps/minute), stride/step length (m) [97,98,99,100,101], distance to base of support (m) [9], support duration (s) and step length variability (mm) [99]), kinematic (midfoot tibial angle [102,103] and hip/knee/ankle absolute angles [101,102]), kinetic (ground reaction force [101,102] and knee/ankle joint torque [102] or pressure distribution in foot sole [104]) characteristics. However, consistent changes in specific gait parameters could not be obtained from the experimental results.

#### 3.4.9. Flooring Type

In this paper, two systematic reviews [38,46] discussed the effects of floor materials or attachments such as compliant floors and carpets on the gait of older people. A compliant floor is broadly defined as a floor system or floor covering with a certain damping capacity. General commercial carpets also belong to compliant floors [46]. The systematic review of Lachance et al. recorded how different types of compliant floors affect the walking balance of older people. Overall, participants were able to maintain static and dynamic balance on carpets and compliant floors. Of the 14 records that looked at gait/mobility measures, 6 did not report any significant changes when walking over carpet or novel compliant flooring (NCF) compared to standard flooring [105,106,107,108,109,110], 5 reported beneficial attributes of carpet or NCF on gait performance [111,112,113,114,115], and 3 reported negative attributes of carpet or NCF on gait performance [116,117,118]. However, there are also studies suggesting that residential and commercial-grade carpets do not have a negative impact on balancing strategies for older people. The researchers conducted a similar study on commercial-grade carpets with poor flexibility and found the same results in older people: The selected carpets did not significantly affect postural swing [38].

A few records suggested some carpets (versus standard flooring) were favorable for walking when gait speed [115], step length [115], walking pattern [112], and obstacle avoidance [113] was considered. Willmott suggested walking on carpet was more efficient than walking on vinyl based on observed increases in gait speed [115]. However, Stephens et al. [117] found carpet resulted in slower walking speed versus parquetry flooring. Walking on rubber flooring (thickness not reported) also resulted in an increase in energy losses. No differences were found in six of the eight gait parameters tested. Only Hanger et al. concluded that the use of low-impact flooring negatively affected gait [118].

## 4. Discussion

In this paper, we aim to summarize the systematic effects of the interventions of the elements of the human settlement pedestrian environment on the gait characteristics of older people and to sort out the mechanisms of environmental factors on the gait of older people, with the ultimate goal of providing evidence for urban builders and researchers. To achieve this goal and to make better use of the data, we analyzed and discussed the available data (see Figure 4).

### 4.1. Affected by the Pedestrian Environment: The Cautious Degree of Active Control Gait

Cautious gait is considered a protective gait pattern [119,120], features include reduced gait speed, cadence, single limb support phase, increased step length, and saccade/step interactions time [7,121,122]. The nine studies included in this paper suggest that stairs or indoor and outdoor steps [41,42], obstacles [41,45], irregular uneven ground [44], path turns during walking [41], mechanical perturbation during gait [40], vibration interventions [40], auditory cues [39], and other factors will influence the degree of caution in older people’s gait.

Previous studies have found that older people have a lower maximum stair height at which they are capable to negotiate [13,14], and need to coordinate greater motor capacity to adapt to stairs [123]. As a result, older people have a more cautious gait profile when walking up stairs [41,42]. This caution increases the sweeping step time during gait initiation [41], and the overall gait cycle becomes longer with reduced stride length and cadence [41,42]. While cautious gait characteristics generally decrease single support time [119] to maintain body stability, the findings support that stair walking increases single support time in older people [41]. This may be due to the longer gait cycle time during stair walking [42], the duration of the single support gait phase is prolonged.

Older people’s ability to cross barriers declines as they age [124], they demonstrated slower crossing speed over an obstacle, reduce step lengths, increased momentum of the foot landing, and a shorter obstacle-heel strike distance [125]. In most of the reviewed articles, older people adopted a slower, more conservative strategy when stepping over obstacles when time was not constrained [41,45], walking slower could provide older people with more time to adjust their foot placement in relation to the obstacle [49,50,51,52,53,54]. Older people judge which environments are challenging pedestrian environments based on their own perceptions. Different kinds of information are required during walking for pre-planning (feed-forward), and for online control (feedback) [126,127]. Pre-planning requires gaze-driven assessment of the environment as a precursor to motor planning and movement execution (e.g., an obstacle within view will contribute toward planning an avoidance maneuver) [41]. This may explain why older people also have longer saccade/stepping interaction time when facing walking obstacles that need to be overcome.

Uneven or irregular ground surfaces can constantly challenge the support and balance of older people when walking. To adapt to changing ground conditions, older people need to constantly adjust their gait and be more cautious [44], exhibiting significantly slower speeds and shorter stride lengths. Whether the reduction of step width here represents a more cautious gait feature remains to be further analyzed. Alfuth et al. believed that this was the stimulus-specific adaptation of gait characteristics caused by plantar feedback changes. However, plantar sensation was not quantified in the study they included but was thought to contribute to gait changes [44]. This may require further research by subsequent investigators on ground conditions and the foot sensory stimuli brought to older people.

Uiga et al. confirmed that turning during walking reduced the pace of older people [41]. On the one hand, older people were found to require more time to scan, perceive, and comprehend the directional changes in their environment when faced with a turn, and to initiate whole-body rotation, as well as head, trunk, and foot repositioning during the subsequent turn [58]. On the other hand, route changes during walking in older people can affect pace rhythm. This has an impact on the kinematic, time–space, and kinetic parameters of walking in older people [41]. They adapted to information loss due to the gradual decline in visual function due to aging, which inevitably slowed their pace [59].

Taylor et al. confirmed that mechanical perturbation during gait can increase the gait cadence of older people and reduce the single support gait cycle [10]. The equilibrium disturbance here refers to the external action imposed on the participants, resulting in the interruption of the balance control. Single support gait is the most unstable phase in the whole gait cycle. It also shows that older people tend to choose to speed up the cadence and control the single support gait cycle time to change their own center of gravity when facing sudden balance disturbance, so as to regain balance [10].

Vibration interventions have improved the gait performance of older people, and vibration interventions have increased the step velocity and walking cadence of older people [40]. Vibration interventions at the threshold affect specific stimuli in older people through tactile and acoustic stimulation. However, the stride length of the older people did not increase significantly, suggesting that vibration interventions were improving gait based on the baseline state of the older people. Stephen et al.’s study provides greater reductions in step variability with the use of vibration for subjects with greater baseline variability and coefficient of variation for step width and stride length increased with stimulation [83]. The more unstable the posture of older people, the higher the sensory feedback threshold [82,88,93,128]. The greater the posture swing of the subjects, the more the vibration noise increases, and the better the gait performance [40].

Ghai et al. demonstrated that for both high and low information processing constraints, cadence and non-rhythmic auditory cues promote athletic adaptation to the environment, and rhythmic auditory cues promote more rhythmic responses to the environment [39]. This also proves the effect of auditory entrainment on the rehabilitation of older people in pedestrian environments. Several mechanisms have been proposed to determine the beneficial effects of rhythmic auditory cues. For example, auditory entrainment may also help to reduce errors during the execution of gait [129,130]. Existing research suggests that the use of ecologically valid acoustic feedback has been suggested [131], and effective movement-related sounds may enhance the salience of sensory information associated with spatiotemporal information, thereby facilitating movement execution [131,132,133,134,135].

Overall, as older people grow older, their physical function and athletic ability decline. Such as decreasing strength and flexibility, visual changes, decreasing balance, decreasing cognitive function, and their physical function and athletic ability decline [43]. This cautious gait may represent an attempt by older people to minimize the speed of the lower extremities and momentum produced during the movement and suggest that older people attempt to minimize the time spent in single support. An increase in momentum may make it more difficult for the older individual to control the lower extremities and increase the chances of a loss of control during the step, single support represents the most unstable period of time during gait and stepping [124]. When older people are faced with the above elements of the pedestrian environment, they actively control their gait in advance, adjusting their body posture and walking strategy to the environment and developing a cautious gait.

### 4.2. Affected by the Pedestrian Environment: Gait Stability Changes Caused by Gait Adaptation

Stability is an important gait performance in older people and a valid predictor of fall risk [136]. Previous studies defined increased gait variability as one of the worst gait changes. In this paper, the center of mass sway [10], pressure center displacement [37,40], margin of stability (MOS) [10], gait variation coefficient [39], posture swing degree [38,40], gait variability [40,44], posture stability [40], dynamic stability [10], dynamic balance [38,40,43], and so on are all used as the characteristic values of gait stability. In the pedestrian environment of human settlements, irregular uneven ground [44], crossing obstacles during walking [37], mechanical perturbation during gait [10], vibration interventions [40], auditory cues [39], and other factors have been found to affect the gait stability of older people.

It was found that to adapt to irregular or uneven ground surfaces, older people constantly adjust their gait and exhibit higher gait variability [137]. This gait adaptation may be a mechanism for increasing postural stability in older people [119,120]. Older people needed longer to terminate walking than younger adults. Delays in the onset of muscle activities, reduced reactions, and a reduced plantar sensation leading to reduced gait stability were discussed in this context. These changes are thought to be a normal mechanism for older people to adapt to changes in their walking environment [44].

Kinematic data from a retrospective study of obstacles in a pedestrian environment show that a cautious gait in older people may be used to improve gait stability when preparing to avoid or step over obstacles [58]. Increased gait variability increases the likelihood that the foot will hit an obstacle while stepping [44], as tripping or falling is often the result of insufficient stride leading to premature ground contact or encountering an obstacle in the path [125,138]. One of the gait strategies noted by the researchers was an increase in the elevation of the swing leg over the obstacle in an effort to maintain a safe clearance distance between the leading foot and the obstacle. In addition, swing velocity (the velocity of the swing leg going over the obstacle) and momentum was decreased in the older participants, which may also be a safety mechanism used by older people because of a loss in upper torso postural control during the movement over the obstacle. Older people adopt this form to allow for increased control of the upper torso during normal walking [10,45,139].

This paper also found that older people will increase the base of support, increase the dynamic stability and reduce the center of gravity swing under mechanical perturbation during gait. However, there are differences in the results of the margin of stability [10]. This discrepancy in the findings for outcome variables was likely a result of the heterogeneity in the choice of perturbation magnitude, gait phase of application, and the measurement approaches used to quantify gait variables. This may also have resulted in the differences in sample population ages, who may have responded differently to different perturbation types. This may be related to the stimulation of foot receptors in older people by mechanical perturbation during gait, which is the internal mechanism of the exercise adaptability of older people in the face of environmental changes.

Costa et al. study showed that noise-based interventions may increase physiologic multiscale complexity and enhance postural stability in older people subjects [140]. In one study, the application of vibration stimulation to the plantar area had a beneficial effect on women aged 60 or over with balance deficits, especially under closed eyes [87]. This is because local vibration stimulation applied to the ankle and foot affects proprioceptive input and balance control. In the closed-eye state, the compensation choice of the subject’s balance control is less. Therefore, the subjects rely more on the residual information source of the mechanical receptors on the sole side of the foot to control their balance. Improving the sensation of plantar surface pressure by vibration stimulation can improve balance when subjects cannot use compensation strategies [128].

Studies have also found that the physical environment may affect gait stability in older people and that physical environment interventions may be effective in reducing risk factors that may contribute to falls in older people and improving postural control [10,40,141]. In this paper, we found that a cadence sound intervention in an acoustic environment can improve walking performance and reduce gait variability in older people by improving their gait cadence [39]. Interventions in the physical environment have an impact on older people’s cognition and understanding of their environment.

The degree of cautious and conservative gait in older people is also found to be positively correlated with gait variability from the evidence plots [10,39,40,44]. Walking on uneven and irregular ground surfaces in older people results in reduced gait speed, cadence, and stride length, and increased stride variability [44]. After the application of rhythmic auditory cues in older people, the step speed, step length, and step frequency all increased, but the coefficient of variability stride time and coefficient of variability stride length decreased [39]. However, when older people walk under the stimulation of vibration, their gait speed and cadence increase, their postural sway decreases, and their dynamic balance increases [40]. Under the action of perturbations during gait, the single support and swing time of the older people is reduced, the cadence is accelerated, and the dynamic balance is increasing [10]. These relationships indirectly demonstrate the link between cautious gait and gait stability. The adoption of a cautious gait in older people aims to enhance walking stability [142,143,144]. A decrease in stride length results in an increase in the number of steps, and although this change results in a decrease in the number of periods of instability during gait, normal and effective gait is compromised. A stiff, slow, and unstable gait can develop [145]. These factors exacerbate the increased gait variability [58].

Overall, on the one hand, these results describe the acute effects of these human settlements’ pedestrian environmental factors on gait, which subsequently lead to changes in gait stability [146]. On the other hand, gait variability is actually a positive reflection of the different system conditions that regulate movement. The human motor system finds the most efficient movement by engaging in a large number of similar motor adaptations in an effort to determine the most efficient and appropriate movement pattern for that particular situation. Indirectly, the interaction of the human homeostatic system and the corresponding environment is demonstrated.

### 4.3. Pedestrian Environment Factors Influencing Other Physical Performance and Control

In this paper, the systematic review of the outcome parameters reveals that in addition to the commonly reported kinematic parameters of older people that reflect changes in cautious gait and gait stability, complex time-space and kinetic parameters represented by joint angles, muscle activation, force moments, and ground reaction forces are also reported. These parameters represent the response of the older people’s body posture and function to the pedestrian environment. In the reported literature, two systematic reviews have shown that crossing obstacles during walking [45] and mechanical perturbation during gait [10] can affect these gait parameters.

In the previous report on obstacle kinematics, older people had more hip bending and adduction when standing [50,147]. An analysis of joint powers indicated that older people demonstrated reduced eccentric contraction of the stance limb hip abductors and vertical hip power (defined as the power used to raise the stance hip) at toe-off were both reduced in older people. Reduced hip abductor strength may lead to pelvic drop during stance, effectively lowering the height of the crossing limb [45]. Another study suggested that older people would opt for an increased base of support and increased muscle activity under mechanical perturbation during gait, but there are inconsistent results about the angular range of joint motion. The large differences in muscle and joint activity results at different sites do not suggest that the effect of mechanical perturbations during gait on muscle activation in older people is controversial and may involve a specific response of older people following physical function aging [10].

This review also observed detailed information on environmental factors that may influence the physical activity of older people during the analysis. Some studies concluded that pedestrian infrastructure, amenities, and environmental conditions will influence the amount of physical activity of older people [148], and the accessibility of the human settlement environment was positively associated with the physical activity of older people [149]. Similar results could be found in a previous study that safe, walkable human settlement areas positively influenced older people’s participation in physical activity, and walkable infrastructure had a positive effect on the amount of physical activity of older people [150]. However, it has also been suggested that the results of the effect of pedestrian environment on the physical activity of older people are inconsistent and more research is necessary in different contexts [5]. The strength of evidence for the association of specific categories of pedestrian environment attributes with physical activity varies depending on the amount of physical activity and type of environmental measurement. Future studies should take note of these findings and identify potential mechanisms.

Overall, the current study suggests that performance in terms of physical function and limb posture, as well as muscle activity, are physiological manifestations of motor adaptations in older people in the face of changes in the pedestrian environment. Although only the parameter evidence of the older people crossing obstacles or experiencing mechanical perturbation during gait has been found in the articles currently included, a possible reason is the lack of systematic review of existing studies, which leads to relevant primary studies not being included.

### 4.4. Disputed Pedestrian Environment Factors Affecting the Gait of the Older People

This paper finds that the impact of human settlements’ pedestrian environmental factors caused by walking in some residential areas is controversial. Among them, the influence of mechanical perturbation during gait [10] and vibration interference [40] on some specific gait parameters of older people are controversial. The effects of texture material [43], floor type [38,46], and other factors on the gait of older people failed to yield consistent results.

There are conflicting results on the effects of mechanical perturbations during gait on step width and length in older people [10]. This may be the result of heterogeneity in the way different experimental participants were subjected to mechanical perturbation during gait. The most common method of balance perturbation in the gait selected here was by treadmill translation, and the conditions of timing, frequency, speed, and intensity of the perturbation performed varied widely. Furthermore, because of the existence of thresholds for the disturbance of the environment that can be perceived by humans, it may have caused different intervention effects triggered by differences in experimental conditions [10].

The intervention methods, frequencies, and characteristics of the effects of the vibration interventions included in this paper also vary. In the pooled analysis, their intervention effects were divided into two parts, considering vibration alone and considering vibration noise, and the interventions were limited to subthreshold vibration interventions. One study showed that vibrating insoles had no effect on balance in healthy older people subjects [147]. The vibratory insoles significantly improved the performance of healthy older people on the Timed Up and Go (TUG) test. The reduction in TUG test times suggests that mobility might be affected [85]. The maximum amount of vibration that could be applied was not sufficient for all participants to reach sensory detection thresholds, which may explain this variation between studies [40].

In this paper, the texture material, compliant flooring systems, and carpet are all pedestrian environment factors that can directly stimulate plantar sensation. The sum of the findings of changes in plantar sensory input in kinematics supports the view that the sensory feedback of the plantar receptors is involved in determining the movement strategy or recovery action during walking. However, combined with the results, the effectiveness of adding plantar stimulation to healthy older people participants is not completely determined. There are still questions about which outcome indicators reflect the improvement of function.

In the conceptual definition of this paper, the morphology of textured surfaces used in previous work has been defined by a huge parameter space including variables such as nodule height, shape, material, area, and packing density [43]. Previous studies have shown that texture can enhance the sensory input from the indentation area. Previous research has reported that textured materials can influence somatosensory system functioning during mechanical interactions with specialized cutaneous receptors [151]. Texture is introduced on the basis that it can stimulate peripheral receptors that are otherwise not being stimulated [43,152], changes in plantar feedback lead to stimulus-specific adaptation of gait features. Increasing the simple mechanical deformation of the texture on the skin surface stimulates the sensory receptors in the skin and improves the function of the perceptual motor system [148,153] facilitating the tighter regulation and control of spatial and temporal characteristics of the center of mass over an individual’s base of support. Improvements in the ability to detect information changes, such as changes in balance [154,155] or the positioning of a limb [156], support adaptive regulation of movement [99,157]. Stimulation of plantar receptors is positively correlated with improvement of motor performance [43].

This paper confirms that the effect of compliant flooring systems on gait and mobility is limited [38,46]. Residential and commercial-grade carpets will not have a negative impact on the balance strategy of older people, and the selected carpet will not significantly affect the posture swing. This may be due to changes in the carpet did not reach the threshold of older people’s gait changes [38]. Available evidence shows that the flexibility of the floor system currently in use has a limited impact on gait and mobility unless the ground hardness is significantly reduced. Perhaps balance and gait will eventually be impaired by a significant reduction in ground hardness, but the specific threshold is related to the individual’s physical condition [46].

In general, the changes in ground type and material did not have a significant unity in specific experimental results. Mechanical perturbation during gait and vibration interventions can affect older people’s understanding and perception of the environment, but it is still controversial to reflect the specific gait effects. This may be due to the large heterogeneity of the ground change stimuli, and the threshold relationship between the foot feeling and the gait change of older people is still uncertain. It is obviously an important research task to find environmental interventions that can provide the greatest benefit for sensorimotor function.

### 4.5. Limitations and Future Research Directions

The findings of this review should be interpreted with consideration of a number of limitations. In this paper, the articles were selected without restricting to specific geographic areas, and this paper provides a more systematic body of evidence on various factors in the pedestrian environment of human settlements. Based on the discussion above, different types of human settlements in different geographic areas can still find the systematic evidence provided in this paper for specific problems in the pedestrian environment, which is the limitation as well as the universal value of this paper. This review does not offer a quantitative summary (i.e., meta-analysis) of the relationships due to the heterogeneity of some studies selected in some of the systematic reviews reviewed, and only a narrative comprehensive analysis is conducted. There is a wide range of quality assessment scores among the reviewed studies, and, in particular, studies that scored low on external validity may limit the universality of the finding. In this paper, we found that the tendency of changes in the cautious gait and gait stability of older people could be stronger with the increase in age and the decrease (weakness) in physical ability [42,44,45]. However, we were not discussing the groups of older people differentiated by age, as there was insufficient experimental data. In order to include as much evidence as possible, the included studies are more tolerant of heterogeneity. We included these studies in the review to enable a comprehensive synthesis of the evidence. Furthermore, the diversity of different health states of the older people study is excluded in order to seek general patterns under normal aging.

In future studies, firstly, not all environmental changes directly trigger physical responses in older people, and the impact of human settlements’ pedestrian environments on gait in older people needs to be further quantified. Future studies should investigate the mechanisms of balance recovery during walking in environments as well as the effectiveness of environmental interventions to mitigate falls and injuries, to quantify the thresholds of the quantitative characteristics of the pedestrian environment (ground flatness, slope, physical characteristics, etc.) that affect the gait change of the older people and provide specific data support for the construction of a pedestrian environment suitable for the older people. Secondly, a more cautious and conservative gait was considered to be one of the gait disorder patterns in older people in previous studies. It is more pronounced or triggered with age and is more dangerous to the health of older people. In contrast, such changes are relatively mild or at a higher threshold in younger people. We will focus on the comparison between older people and young people in terms of specific influencing factors, influencing effects, harmfulness, and the specific effects of these factors on the walking behavior of older people in different age groups in future studies. Thirdly, older people will not only cause real-time gait changes when walking in the human settlements pedestrian environment. Attention should also be paid to issues related to postural impairment and the functional load of older people when walking in the environment for long periods of time. The correlation between characteristics of the ground-based pedestrian environment and long-term trained gait parameters should be further explored to investigate the long-term effects of the pedestrian environment on older people when used as a gait intervention variable. Fourthly, an association between changes in gait variability and fall risk in older people was observed in the study, but whether this correlation can be used to directly quantify fall risk in older people in the environment remains uncertain and the issue warrants in-depth study. Moreover, as part of the daily physical activity of older people, pedestrian environmental factors that influence older people’s walking behavior also have a significant impact on physical activity. The amount of physical activity of older people is closely related to their physical health. It is worthwhile to focus on the issue of how the amount of physical activity of older people may change when changes in their gait characteristics occur due to the living environment.

## 5. Conclusions

This paper is an umbrella review aimed at synthesizing the systematic effects of the human settlement pedestrian environment on gait in older people. This paper comprehensively reviews the detailed evidence of the current 11 systematic reviews on the intervention of pedestrian environment in human settlements on the gait of older people. We found that the main human settlements pedestrian environment factors that can cause gait changes in older people are indoor and outdoor stairs/steps, uneven and irregular ground surfaces, ground obstacles, path turns during walking, vibration interventions, mechanical perturbations during gait, and auditory cues. Under the influence of the pedestrian environment described above, older people will mainly experience changes in the degree of cautious conservatism and stability of their gait, and their body posture performance and control, and muscle activation will also be affected. Factors such as ground textures or materials, vibration interventions, and mechanical perturbations during gait stimulate older people’s understanding and perception of the environment, but there is still controversy on the specific gait impact. In view of this, we come to a unanimous conclusion that human settlements’ pedestrian environment affects the gait changes of older people in a positive or negative way, as a manifestation of their motor adaptations to the environment after perceiving environmental stimuli while walking. Based on the findings of this paper, it is recommended that architects and urban planners should take into account the creation of appropriate pedestrian environments for older people when constructing human settlements and housing in order to improve gait performance and reduce safety risks for older people in pedestrian environments. This review may likely contribute evidence-based information to aid communication among practitioners in public health, healthcare, and environmental construction. It may also provide a basis for future research on novel environmental interventions to improve the walking health and safety of older people. The above findings are expected to provide useful preference for associated interdisciplinary researchers to understand the interactions among pedestrian environments, human behavior, and physiological characteristics.

## Figures and Tables

**Figure 1 ijerph-20-01567-f001:**
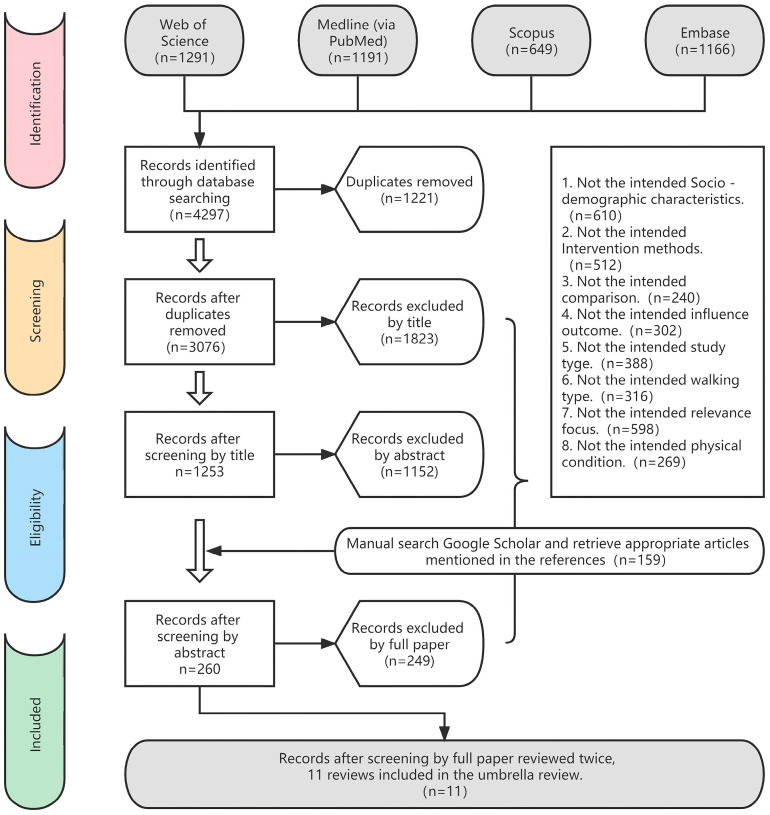
The procedure was followed using the PRISMA flowchart.

**Figure 2 ijerph-20-01567-f002:**
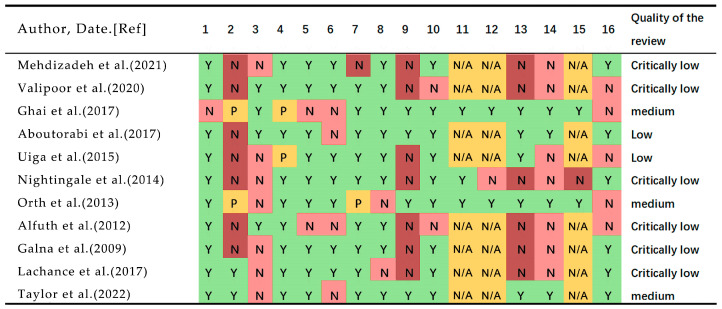
Quality assessment of the included systematic reviews using the AMSTAR 2 checklist [10,37,38,39,40,41,42,43,44,45,46]. (This relative quality assessment tool follows the AMSTAR2 checklist [30].) This scale has four ratings for systematic reviews: critically low, low, moderate, and high; Y: yes, P: partial yes, N/A: not applicable, N: no; green: no flaw, light red: non-critical flaw, deep red: critical flaw, yellow: partial yes or not applicable.

**Figure 3 ijerph-20-01567-f003:**
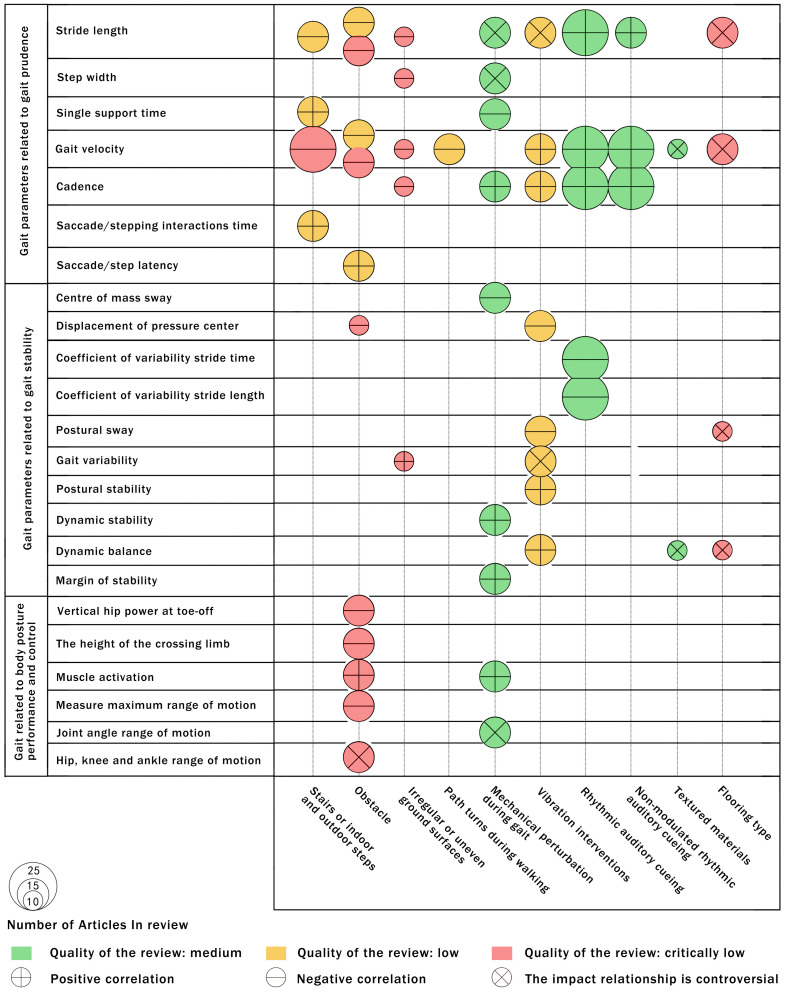
Descriptive evidence map of umbrella comprehensive review. (On the x-axis, studies were classified according to the environmental variables studied, while on the y-axis, they were classified according to the gait characteristics of the older people studied, while the quality of the studies and the number of samples systematically reviewed were differentiated for each pair of role relationships. Study quality was classified as medium, low, and very low (according to the results obtained from AMSTAR 2) and distinguished by color, and the number of studies systematically reviewed per study was classified as 1–10, 11–15, and 16–25, distinguished by the size of the graph. The relationship between the role of environmental elements and gait characteristics of older people was also distinguished in three forms: positive correlation, negative correlation, and presence of controversy).

**Figure 4 ijerph-20-01567-f004:**
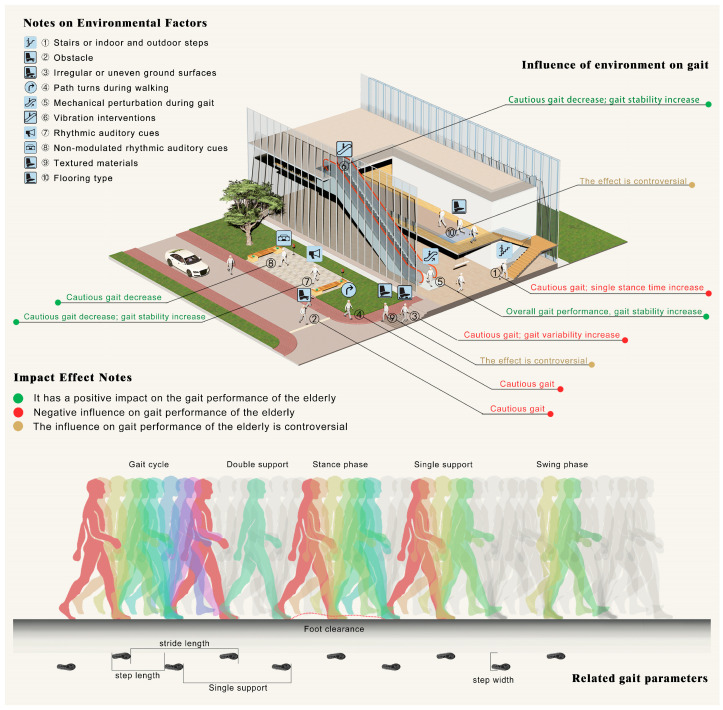
The influence of human settlements environment on the gait of the older people and the description of relevant gait parameters.

**Table 1 ijerph-20-01567-t001:** PICOT (population, intervention, comparison, outcome, type of study) scheme to define inclusion criteria.

Parameter	Description	
	Inclusion	Exclusion
Population ^1^	Contains at least some subgroups of study subjects who are older people, non-institutional self-sufficient citizens, older people without definite physical illness with no reported disease, mean age > 60 years, or authors claiming that the study subjects are older people.	Older people with significant physical health problems, children, adolescents, young adults, institutionalized/hospitalized.
Intervention	Short-term interventions for walking in the environment, characteristics of the built environment: physical environment, visual environment, auditory environment, gait barriers, accessibility, sidewalk barriers, balance barriers, etc.	Long-term training of walking, intervention with walking aids, exercise, drug intervention, walking without environmental intervention.
Comparison	Walking state in natural state (gait and physiological indicators) of older people, baseline physiological state without other intervention in natural state of older people.	Comparison with young people, compare with the gait indexes of older people with various diseases.
Outcome	The influence of human settlements’ pedestrian environment on gait and physiological characteristics of older people.	Studies investigating training, exercise, therapy, and drugs to predict disease risk.
Study design	Systematic/narrative review and/or meta-analysis design.	Narrative review without systematic review

^1^ Self-sufficient, independent older population was chosen because of the huge burden of disease and disability in older people’s health care for health systems. Papers that specifically considered effects of built environment on physical activity in other subgroups were excluded.

**Table 3 ijerph-20-01567-t003:** Effect of intervention measures on gait characteristics.

		Stairs or Indoor and Outdoor Steps	Obstacle	Uneven Surface and Irregular Surface	Turning around a Corner	Mechanical Perturbation during Gait	Vibration Interventions	Rhythmic Auditory Cues	Non-Modulated Rhythmic Auditory Cues	Textured Materials	Flooring Type
Gait parameters related to gait prudence	Stride length	−[41]	−[41,45]	−[44]		/[10]	/[40]	+[39]	+[39]		/[46]
	Step width			−[44]		/[10]					
	Single support time	+[41]				−[10]					
	Gait velocity	−[42]	−[41,45]	−[44]	−[41]		+[40]	+[39]	+[39]	/[43]	/[46]
	Cadence			−[44]		+[10]	+[40]	+[39]	+[39]		
	Saccade/step interactions time	+[41]									
	Saccade/step latency		+[41]								
Gait parameters related to gait stability	Centre of mass sway					−[10]					
	Displacement of pressure center		−[37]				+[40]				
	Coefficient of variability stride time							−[39]			
	Coefficient of variability stride length							−[39]			
	Postural sway						−[40]				/[38]
	Gait variability			+[44]			/[40]				
	Postural stability						+[40]				
	Dynamic stability					+[10]					
	Dynamic balance						+[40]			/[43]	/[38]
	Margin of stability					+[10]					
Parameters related to attitude expression and control	Vertical hip power at toe-off		−[45]								
	The height of the crossing limb		−[45]								
	Muscle activation		+[45]			+[10]					
	Measure maximum range of motion		−[45]								
	Joint angle range of motion					/[10]					
	Hip, knee, and ankle range of motion		/[45]								

−, Negative correlation; +, Positive correlation; /, Correlation unclear or disputed; Gait speed: Distance walked per unit time; Step width: Lateral distance between the heels for consecutive heel strikes of the two feet; Stride width: Perpendicular distance from the heel of one foot to the line connecting two consecutive heel strikes of the contralateral foot; Step length: Distance between the heels in the anteroposterior direction for consecutive heel strikes of opposite feet; Stride length: Distance between the heels in the anteroposterior direction for consecutive heel strikes of the same foot; Stride time: Time between consecutive heel strikes of the same foot; Cadence: Number of steps taking per unit time; Gait variability: Step-to-step deviations/variations in gait parameters; Gait cycle: One complete cycle of one limb starting when the foot first contacts the ground to when the same foot next contacts the ground; Stance phase: Phase of gait cycle from touchdown to liftoff of the same foot; Swing phase: Phase of gait cycle during which the foot of interest is not on the ground; Double support: Both feet are simultaneously contacting the ground; Single support: Only one foot is in contact with the ground.

## Data Availability

No new data were created or analyzed in this study. Data sharing is not applicable to this article.

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
