# Peer review of "The Effect of Human Settlement Pedestrian Environment on Gait of Older People: An Umbrella Review"

_ijerph, 2023, doi:10.3390/ijerph20021567_

Round 1

Reviewer 1 Report

Dear authors, thank you for the opportunity to study your text. It is a well-crafted study and its contribution can reach different fields of research. I have a few rather technical remarks, please consider them and incorporate them as you find fit.
 In the introduction you talk about the aging of the population, but it is not clear what relevance this has to your text, which deals with current (or even past) generations of older people. I know that this is a common opening to texts dealing with the older people, but I feel that it is used without reflection, to the point of being overused and without relevance to the actual message of the text. In the case of your text, the informational value would not be diminished in any way if you delete this paragraph smacking of demographic panic. It can be replaced with a more general statement of why the topic you are focusing on, your research question, is relevant. The information is repeated several times as to who it may be relevant to (urban planners etc.), but it is not entirely obvious to the interdisciplinary readership "what your conclusions are good for", what they are useful for. To oversimplify, for which I apologize, the result is that seniors are walking more slowly and more cautiously. But then again, so do younger people if they hit an uneven surface or obstacle. So maybe the difference is in the scale not in the (non)existence of the phenomenon. In what sense, then, is our knowledge moving forward? At the same time, please consider how to achieve greater cultural sensitivity. The general definition of "human settlement pedestrian environment" can include both Sub-Saharan Africa and downtown New York.

One possible solution is reflection in the broader context of selected social science disciplines (at the moment the findings speak mainly to a specific group of professionals for whom the results are probably not surprising/unknown, but can be placed in contexts for which they have significant innovative potential), recognition of the heterogeneity of the senior population (who are the seniors in the studies reviewed ? 60+/80+), and finally defining the frame of reference: are the results relevant for frail seniors versus non-frail, seniors versus younger people? I may have missed it, but were the interventions conducted in a natural setting or in a laboratory? Did it affect the results in any way?

Technically, please change the term "elderly" to "older people" (see the 1996 UN recommendation).
On line 253, please modify "naturally aging elderly people", the implication that there are "un-naturally aging people" does not seem appropriate to me; are these people living in natural environments (as opposed to nursing homes?) or people who are aging in a specific way?
Please make sure to explain all abbreviations before their first appearance in the text, and/or make sure that their use is necessary (e.g. sections 322 - 327); the interdisciplinary readership will welcome this;
In section 159 and following, insert a space before the left parenthesis.
Can the term "disorder" be replaced by some other word? It may be only my understanding, but it evokes something disordered to me, rather than, for example, some physical obstacle on the sidewalk.

Notwithstanding the above, I believe that the text, with some minor editing and reframing, will make an interesting contribution to the argument about the importance of the physical environment to quality of life in older age and will find readers in a variety of disciplines. Technically, the text is correct, clear and with a high attention to detail. Thank you once again for allowing me to read it.

Reviewer 2 Report

This paper aims to examine the systematic effects of the pedestrian environment on older adults' walking from multiple perspectives and comprehensively. The design of the study is an umbrella review.

Its methodology is consistent and clearly stated. In an umbrella review, the method of selecting systematic reviews for inclusion in the study and the inclusion/exclusion criteria are fundamental aspects, which are also clearly presented.

I recognize that this paper could be a significant one that provides direction in developing housing environments for the elderly.

Major Comments

The notations in the paper are transparent and allow the reader to understand the research thoroughly. In particular, I recognize that what is shown in Figure 3 is a significant figure that covers the critical visual points and aids in understanding the research.

If possible, I believe that the critical meaning of Figure 3 could be clarified by splitting it into two figures, one for the living environment and the other for the gait phase. I recognize that Figure 3 is significant in this paper, and please consider splitting the figure for better understanding.

A complete discussion based on the study results has been advanced in the DISCUSSION part of this paper. I am fully aware that this study focuses on the gait characteristics of the elderly and the mechanisms by which the environment affects the elderly's gait.

However, I believe that one perspective that should be considered, in part, is a discussion of how the amount of physical activity of the elderly may change when changes in their gait characteristics occur due to the living environment.

In limitation, I believe that the next issue, as the author states, is to examine thresholds related to each living environment. Finally, I think the amount of physical activity should also be considered, and please consider considering the impact on the amount of physical activity of the elderly.
